# Profile of Selected Mineral Elements in Tibiotarsal Bone of the White-Tailed Sea Eagle in Its Natural Habitat

**DOI:** 10.3390/ani12202744

**Published:** 2022-10-13

**Authors:** Jakub J. Ruszkowski, Anetta Hanć, Marcin Komosa, Małgorzata Dzierzęcka, Tadeusz Mizera, Maciej Gogulski, Anita Zaworska-Zakrzewska

**Affiliations:** 1Department of Animal Anatomy, Poznan University of Life Sciences, Wojska Polskiego 71C, 60-625 Poznań, Poland; 2Department of Trace Analysis, Adam Mickiewicz University, Uniwersytetu Poznańskiego 8, 61-614 Poznań, Poland; 3Department of Animal Physiology and Physiotherapy, Bydgoszcz University of Science and Technology, Mazowiecka 28, 85-084 Bydgoszcz, Poland; 4Department of Morphological Science, Faculty of Veterinary Medicine, Warsaw University of Life Sciences, Nowoursynowska 166, 02-776 Warsaw, Poland; 5Department of Zoology, Poznan University of Life Sciences, Wojska Polskiego 71C, 60-625 Poznań, Poland; 6Department of Preclinical Sciences and Infectious Diseases, Faculty of Veterinary Medicine and Animal Science, Poznań University of Life Sciences, 60-637 Poznań, Poland; 7Centre of Biosciences, Institute of Animal Physiology, Soltesovej 4-6, 040-01 Kosice, Slovakia; 8Department of Animal Nutrition, Poznan University of Life Sciences, Wołyńska 33, 60-637 Poznań, Poland

**Keywords:** bone composition, conservation, heavy metal toxicity, minerals, Pb toxicity, *Haliaeetus albicilla*

## Abstract

**Simple Summary:**

Analyses in this paper are concerned with one of the biggest avian raptors distributed widely across Eurasian territory. The White-Tailed Sea Eagle is considered one of the most important protected species of wetlands including plains and seashores. Monitoring heavy metal toxicity in raptors may contribute to the prevention of intoxication in other species, including humans. Researching their biology and different environmental connections may help protect the whole habitat. In the paper, the results of the analysis of the mineral composition of long bones collected from seven White-Tailed Sea Eagles are presented. Specimens came from the territories of Western Poland. It is the first paper describing the bone composition of this species.

**Abstract:**

Mineral bone composition (dry matter, bones ash, P, Ca, Zn, Mn, Mg, and Cu) and Pb levels of tibiotarsi of seven White-Tailed Sea Eagles were assessed. Lead intoxication in different bird species including waterfowl and raptors is being studied worldwide. The bones were analyzed for Pb by mass spectrometry with excitation in inductively coupled plasma (ICP-MS Elan DRC II) and for bone composition by Atomic Emission Spectrometer (Agilent 4100 Microwave Plasma). Pb levels ranging from 3.54 µg/g to 74.6 µg/g DM suggest that some of the investigated birds might have been intoxicated by Pb. Results of this analysis were divided into two groups of bones, with bone Pb levels higher and lower than Pb toxicity levels, and mineral bone compositions of both groups were compared. The present study shows the differentiation of bone mineral composition among seven examined White-Tailed Sea Eagles, considered a specific species in raptors. Pb intoxication may not have a major influence on mineral bone composition in raptors. It also suggests that assessing bone composition of raptor bones may help finding the possible cause of their deaths.

## 1. Introduction

Pb intoxication as a result of ingested Pb ammunition or its particles in birds has been subject of research for years. The problem is being studied worldwide in different bird species including waterfowl and raptors, the two most exposed groups of birds because the waterfowl is hunted in many countries, and it is also raptors’ prey. Waterfowl are becoming intoxicated with Pb by unintentional ingestion of spare Pb gunshots while feeding. Birds of prey feed on animals that have been shot or feed on gut piles left after hunting by man. In many countries, there have been numerous limitations regarding the usage of Pb ammunition in hunting waterfowl in order to reduce the amount of Pb introduced to the environment [1,2,3]. Such regulations may contribute to decreasing amount of ways birds can become intoxicated with lead [4]. In Poland, lead shots are the major kind of ammunition used by hunters. 

Birds of prey are a large group of birds that prey on other animals or their carcasses. Being on the top of the food chain, they easily become intoxicated with chemicals accumulated in their preys’ bodies [5,6]. Many of them also die after a collision with transmission lines, wind farms, or trains, where their cadavers may be found [7,8]. The White-Tailed Sea Eagle (*Haliaeetus albicilla*) is one of the biggest raptors distributed widely across the whole of Eurasia and is considered one of the most important umbrella species of European wetlands. Monitoring heavy metal toxicity in raptors may contribute to the prevention of intoxication in other species, including humans [9]. Researching their biology and different environmental connections may help protect the whole habitat. Their diet and feeding behavior change seasonally. In spring and summer, the diet includes mainly fish, in fall waterfowl, and in winter mammals, usually obtained as carcasses or piles of gut left by hunters after eviscerating prey [6,10,11]. However, the major cause of death in White-Tailed Sea Eagles is Pb intoxication due to the ingestion of lead bullets or their particles. There are also reports of eagles being shot with Pb ammunition [7].

The bone tissue that builds the skeleton of a bird consists of both organic and inorganic components. Previous studies described levels of dry matter, ash in dry matter, and other elements in the bones of many bird species, including poultry, psittacines, and some raptor species [12]. According to the literature data, in poultry, dry matter ranges from 45% to 88%, while in White-Tailed Sea Eagles from 80.5% to 91.2% (studies were carried out on 13 eagles) [12]. The skeleton consists mainly of calcium and phosphorus in a ratio of 2:1, and the calcium content of the bones increases with age in poultry. The phosphorus levels are not consistent. In different bird species the ratio ranges from 1.9:1, in poultry and various psittacines, to 2.5:1, in budgerigars. In birds of prey, as a single paper shows, calcium contents in a single buzzard and a single falcon are 325 g/kg and 329 g/kg, respectively [12]. About 60% of the total magnesium amount in the body is stored in the bones. In one study with raptor species, the levels were 4700 mg/kg ash magnesium in one falcon and 6100 mg/kg ash magnesium in one buzzard, which is higher than median magnesium levels in other described bird species [12].

Zinc level is related to bone growth since zinc is an enzyme cofactor important for bone metabolism. In humans, zinc deficiency is related to osteoporotic lesions [13]. Copper is an essential trace mineral that has been found as an important factor of bone metabolism in mammals. The role of copper in bone formation and mineralization is yet not fully understood. It is an important enzyme cofactor in different bone-related reactions [13]. It is a constituent of some enzymes and activator of other enzymes. Tissue levels of manganese can be related to the condition of the bone and cartilage tissue due to interactions between osteoblastic and osteoclastic activity [13]. Moreover, zinc, copper, and manganese levels may influence calcium metabolism [13].

The aim of the present study was to assess element composition (calcium, phosphorus, magnesium, zinc, and copper) and Pb levels of seven individual White-Tailed Sea Eagles. Moreover, the frequency of Pb intoxication incidents of White-Tailed Sea Eagles in the Western Poland and the influence of Pb levels on mineral bone composition were analyzed.

## 2. Materials and Methods

### 2.1. Sample Collection and Preparation

The bones were collected post-mortem from eagles found dead in Western Poland. The eagles were divided into two age groups based on professional ornithologist assessment, subadult (3–4 years old) and adult (over 6 years old). Because three eagles had been ringed, it was possible to assess their exact age. Individuals were numbered one to seven. Bird number 1 (subadult male, 4 years old) was found dead near the rail tracks. Number 2 was a subadult male. Number 3 was an adult female. Number 4 was an adult (11 years old) of unknown sex. Number 5 was an adult male. Number 6 was an adult (16 years old) of unknown sex. Eagles numbered 4, 5, and 6 might have died severely weakened since they were found dead on the ground. Number 7 was an adult of unknown sex found near the road, heavily weakened with signs of a car accident (broken bones, several wounds).

From each individual, tibiotarsus was collected (Figure 1). Fresh bones were cleaned from a foreign debris. The bones were boiled to remove their tissues and cartilage caps. Ground dried bone samples were extracted to remove fat. Subsequently, the bone materials were dried at 40 °C in an oven to constant biomass. Dried bones were ground in a ball mill. The bones were analyzed in duplicate to estimate the content of dry matter, using methods 934.01, according to AOAC [14]. The extracted samples were burned in a muffle furnace (P330, Nabertherm GmbH, Lilienthal, Germany) at 600 °C for 5 h, and the ash weight of each sample was expressed as a percentage of the dry bone weight.

### 2.2. Chemical Analysis

Pb was detected by ICP-MS Elan DRC II (PerkinElmer SCIEX, Toronto, ON, Canada). The instrumental parameters used for the determination of Pb were as follows: RF power 1.1–1.2 kW, plasma gas (argon) flow 12 L min^−1^, and nebulizer gas (argon) flow from 0.8 L min^−1^ to 0.9 L min^−1^. The content of P Ca, Zn, Mn, Mg, and Cu was determined according to the procedure of AOAC [14]. The minerals concentrations were measured by Atomic Emission Spectrometer (Agilent 4100 Microwave Plasma Santa Clara, CA, USA). The bone ash material was digested in a microwave digestion system. For the sample digestion procedure 65% HNO_3_ (Suprapur, Merck, Darmstadt, Germany) and 30% H_2_O_2_ (for trace analysis, Sigma-Aldrich, Schnelldorf, Germany) were used. Calibration standards of analyzed elements were prepared by an appropriate dilution of a 1000 mg L^−1^ stock solution of elements in 2% HNO_3_ (Pure Single-Element Standard, Atomic Spectroscopy Standard, PerkinElmer Pure) and a 10 mg L^−1^ multielement stock solution in 5% HNO_3_ (Multielement Calibration Standard 3, Atomic Spectroscopy Standard, PerkinElmer Pure). Analyses of Pb were performed using 159 Tb as internal standards in order to eliminate the drift of the instrument and non-spectral interferences. 

### 2.3. Interpretation of the Pb Level

Pb levels in bone tissue were evaluated with reference to suggested interpretation table for sub-clinical, clinical, and severe Pb poisoning with values of Pb concentration in blood, liver, kidney, and bone tissue published earlier [15]. According to the previous study, sub-clinical poisoning begins at 10 to 20 µg/g, clinical at >20 µg/g of dry matter, and severe intoxication cannot be differed from clinical by Pb level in bone tissue [16]. According to the scheme mentioned above the bones were divided into two groups: group 1 with bones with Pb levels lower than 10 µg/g, and group 2 with increased Pb levels (over 10 µg/g).

### 2.4. Statistical Analysis

All data were analyzed using Statistics v.13.4. (Statistica, StatSoft, Tulsa, OK, USA). The significance of differences was assessed based on the Welch’s t-test with the assumed significance level at *p* ≤ 0.05. The correlation between various parameters was evaluated by Pearson’s correlation test.

## 3. Results

Results were divided into two groups according to bone Pb level. The values for all parameters for both group 1 and group 2 are displayed in Table 1. The mean dry matter percentage was 85.75% (±0.59) for group 1, and 85.8 (±1.54) for group 2. Mean Ca and P levels were 356.64 g/kg (±6.2) and 165.45 g/kg (±1.67) for group 1, and 368.63 g/kg (±3.92) and 165.78 (±3.34) for group 2. Ca to P was 1.7:1 for both groups.

There was also correlation (*p* = 0.67) between contents of Pb and Mn in evaluated bones, which shows that with increase in the Pb concentration level, increase in Mn concentration level in bone tissue is observed (Table 2).

Pb levels were elevated (higher than 10 µg/g) in 4 of 7 individuals, which is more than 57.1% of the researched group. The Pb levels were higher than 20 µg/g meaning clinical or severe Pb poisoning in 3 of 7 and one individual was close to 20 µg/g. Eagle number 4 had levels of Pb in bone tissue reaching almost 75 µg/g. All birds with Pb levels higher than 10 µg/g were adult. Average bone lead level in subadult birds (2 individuals) was 5.34 µg/g and in adult birds (5 individuals, two with precisely known age of 11 and 16) 38.84 µg/g. The Pb levels for each individual bone are shown in Table 3.

The analysis of the two groups showed no significant differences in bone composition in both groups, but there were two minor tendencies. Magnesium and phosphorus levels were lower in bones with increased Pb levels. Figure 2 shows small metallic opacities suggestive for ingested Pb ammunition particles.

## 4. Discussion

Many factors may influence the mineral bone composition of a raptor. Determination of the mineral composition of the bone may be helpful for obtaining more information about animals found dead with an unknown cause. With increased amount of research on the mineral composition of the skeleton and more data about different species and individuals, researchers can build a database helpful in evaluating age, sex, health status, and even a cause of birds’ death. 

In a previous study, the dry matter percentage in long bones of White-Tailed Sea Eagles and two other raptor species was higher than the dry matter percentage in broiler chickens (89.5% for White-Tailed Sea Eagle, 87.6% for common buzzard (*Buteo buteo*), and 88.3% for barn owl (*Tyto alba*) [12], compared with 45–48% for broiler chickens [17]). In the study on African Black Ostriches dry matter of tibiae was 81.13% [18]. This study contributes to these data with similar results for dry matter and bone ash percentage for the eagle. Because of bone mineralization in juvenile birds, dry matter percentage increases with age. The differences between femur and tibiotarsus dry matter percentage can be related to the size and role of the bones. Average calcium and phosphorus levels for long bones in this study were moderately higher than in the study by Schuhmann et al., [12]. The content of Ca was approx. 332 g/kg, P was approx. 196 g/kg; however, Schuhmann et al., [12] found 225 g/kg and 112 g/kg, respectively, for humerus and 235 g/kg and 110 g/kg, respectively, for tibiotarsus. While the previously reported ratio of calcium to phosphorus in raptors was equal 2:1 to 2.3:1 [12], in this study it was slightly lower with the value of 1.7:1.

Pb toxicity in raptors is a serious and widespread issue in conservation medicine worldwide. There is a need for more research on a larger group of animals with more detailed data such as necropsy reports, approximate date of death, and data about hunting in areas where White-Tailed Sea Eagles live since breeding season and birds’ increased activity overlap with hunting season in many countries. In other species, elevated blood Pb influences the skeleton in various ways. It causes an increased frequency of fractures [19] and Pb intoxication is related to a decrease in bone mineralization [20]. Pb also substitutes calcium in hydroxyapatite mineral [21]. This fact may explain differences in calcium to phosphorus ratio in this study and previous studies. Kelly and Kelly [22] described a higher frequency of power lines collision in Mute Swans (*Cygnus olor*) with increased Pb blood levels. 

Our study is the first to describe the bone tissue in the White-Tailed Sea Eagle from Poland and the first to compare bone composition in birds intoxicated with Pb since previous research was conducted on soft tissue samples [5,23]. Two previous studies undertaken by Kitowski et al. [5] and Falandysz et al. [23], showed elevated levels of Pb in soft tissue samples from White-Tailed Sea Eagles’ kidneys and livers. Similar research was conducted in Sweden [24]. Taggart et al. [25] assessed Pb levels in the liver and bones and compiled data about Eurasian Buzzards (*Buteo buteo*) and hunting seasons for gamebirds in the United Kingdom. Feather testing is also a known method for monitoring heavy metal toxicity in different bird species [26,27]. Pain et al. [28] found elevated Pb levels in bone and feather samples of the Spanish imperial eagle (*Aquila adalberti*). Wayland et al. [29] analyzed soft tissue and bone tissue samples from two species of eagles, bald, and golden. The study showed higher concentration of Pb in bone samples in birds that had higher liver Pb concentration > 6 µg/g. The same issue was described in different raptors and waterfowl species [30,31]. Experimental data suggest that bone Pb concentration may be helpful in chronic Pb intoxication research since the concentration in bone tissue raises with age [31], while soft tissue expresses recent exposure better [30]. Results of a study by Taggart et al. [25] where Pb levels in livers and bones were analyzed contribute to these data. Different bone Pb concentrations in different raptor species may reflect numerous dietary habits of birds [32]. The problem of Pb as an environmental pollutant is a worldwide issue. In this study, authors extended the geographical area of the problem by data from western Poland.

## 5. Conclusions

Assessing bone composition and heavy metal levels in bone tissue of dead animals may be helpful in revealing the cause of death. It is also a useful tool to monitor the environment in search for heavy metal pollution, a growing problem worldwide. Collecting data about intoxicated species on top of the food chain may be reflecting the pollution in the whole habitat and be a useful tool in the prevention of heavy metal toxicity to other species, including humans.

## Figures and Tables

**Figure 1 animals-12-02744-f001:**
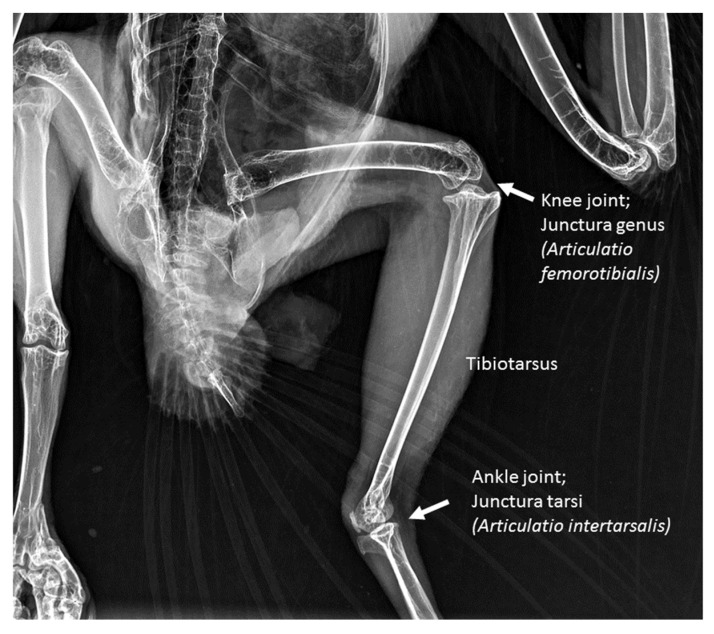
Tibiotarsal bones were dissected in joints pointed by arrows.

**Figure 2 animals-12-02744-f002:**
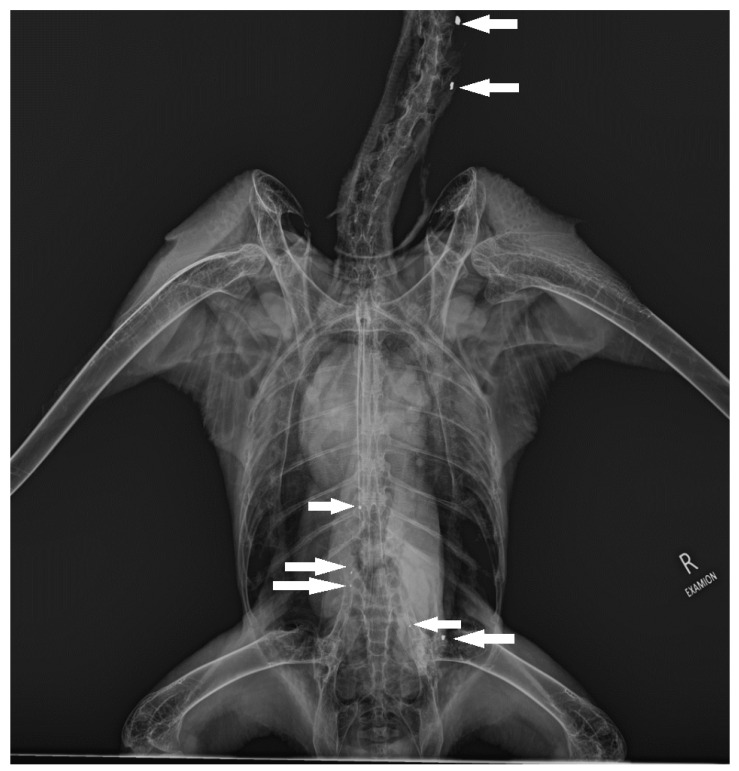
The ventrodorsal radiograph of adult White-Tailed Sea Eagle (*Haliaeetus albicilla*) (number 5). Arrow showing small metallic opacities suggestive for ingested Pb ammunition particles in different segments of gastrointestinal tract.

**Table 1 animals-12-02744-t001:** Means, minimum, maximum, and standard deviation (SD) values for the bone composition in the tibiotarsi of the White-Tailed Sea Eagles with bone Pb level lower than 10 µg/g and higher than 10 µg/g and differences between means (*p*-value).

Parameter	Bone Pb Level Lower than 10 µg/g (n = 3)		Bone Pb Level Higher than 10 µg/g (n = 4)		
	Mean (Min-Max)	SD	Mean (Min-Max)	SD	*p-*Value
DM, %	85.75 (85.22–86.38)	0.59	85.80 (84.41–87.42)	1.54	0.961
Bone ash in DM, %	60.82 (59.60–61.82)	1.13	60.0 (58.17–61.69)	1.47	0.482
P, g/kg DM	165.45 (163.95–167.25)	1.67	165.78 (162.65–169.45)	3.34	0.873
Ca, g/kg DM	356.64 (353.18–360.32)	3.58	368.63 (363.01–372.10)	3.92	0.500
Zn, µg/g DM	26.88 (22.64–30.34)	3.91	26.82 (24.48–30.05)	2.46	0.985
Mn, µg/g DM	8.78 (7.97–9.69)	0.87	11.14 (8.63–14.47)	2.53	0.990
Mg, µg/g DM	3881.64 (3696.54–4055.5)	17.98	3886.32 (3100.75–4561.27)	59.97	0.187
Cu, µg/g DM	2.0 (1.42–2.70)	0.65	2.55 (2.35–2.74)	0.18	0.163
Pb, µg/g DM	4.81 (3.54–7.13)	2.01	45.82 (19.74–74.59)	28.9	0.062
Ca:P ratio	1.7:1		1.7:1		

DM—dry matter.

**Table 2 animals-12-02744-t002:** Correlation coefficient (n = 7) (casewise deletion of missing data), marked values are significant at *p* ≤ 0.05.

	DM_%	Ash_%_DM	P, g/kg DM	Ca, g/kg DM	Zn, μg/g DM	Mg, μg/g DM	Mn, μg/g DM	Cu, μg/g DM	Pb, μg/g DM
DM_%	-	0.75	0.75	0.75	0.75	−0.51	0.44	0.17	0.11
Ash_%_DM	0.75	-	1	1	1	−0.03	0.26	−0.2	−0.25
P, g/kg DM	0.75	1	-	1	1	−0.03	0.26	−0.2	−0.25
Ca, g/kg DM	0.75	1	1	-	1	−0.03	0.26	−0.2	−0.25
Zn, μg/g DM	0.75	1	1	1	-	−0.03	0.26	−0.2	−0.25
Mg, μg/g DM	−0.51	−0.03	−0.03	−0.03	−0.03	-	0.18	−0.03	0.09
M, μg/g DM	0.44	0.26	0.26	0.26	0.26	0.18	-	0.4	0.58
Cu, μg/g DM	0.17	−0.2	−0.2	−0.2	−0.2	−0.03	0.4	-	0.26
P, μg/g DM	0.11	−0.25	−0.25	−0.25	−0.25	0.09	0.58	0.26	-

DM—dry matter.

**Table 3 animals-12-02744-t003:** Detailed bone Pb levels in tibiotarsi of seven White-Tailed Sea Eagles used for analysis.

Bird, No.	1	2	3	4	5	6	7
Age	Subadult, 4 years	Subadult	Adult	Adult, 11 years	Adult	Adult, 16 years	Adult
Sex	Male	Male	Female	Unknown	Male	Male	Unknown
Pb, µg/g DM	3.54	7.13	3.77	74.59	22.18	19.74	66.77

DM—dry matter.

## Data Availability

Data is available at a reasonable request to the corresponding authors.

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
