# Peer review of "Profile of Selected Mineral Elements in Tibiotarsal Bone of the White-Tailed Sea Eagle in Its Natural Habitat"

_animals, 2022, doi:10.3390/ani12202744_

Round 1
Reviewer 1 Report
The purpose of the reviewed work was to analyze and evaluate the comparison of selected mineral elements in tibiotarsal bone one of the biggest avian raptors (White-Tailed Sea Eagle) distributed widely across Eurasian territory. The results of the analysis of the mineral composition of long bones collected from seven White tailed sea eagles are presented and the specimens came from the territories of Western Poland. The authors attempt to show that Pb intoxication may not affect bone mineral composition in raptors, and that assessing the composition of their bones can help determine the the possible cause of their death. The work is interesting, but the authors did not avoid some deficiencies. The determination of the origin of the birds should be placed in the materials and methods, not in the discussion. Moreover, in my opinion, not enough birds were selected for the study. Indeed, determining the mineral composition of bones can be helpful in obtaining more information about birds found dead with an unknown cause, but not in such a small research group. The value of the work would significantly increase if the determinations of elemental content also included organs such as the liver and kidneys. Despite some shortcomings, the topic of the article is adequate to its content. The article is written in a clear and understandable manner. In my opinion, the manuscript can be ready for publication after making the recommended corrections.
Author Response
Dear Editor and Reviewer of Animals,
Thank you for the valuable suggestions and comments made by the editor and reviewers for the improvement of our manuscript entitled “Profile of Selected Mineral Elements in Tibiotarsal bone of the White-Tailed Sea Eagle in its natural habitat ” by Jakub J. Ruszkowski, Anetta Hanć, Marcin Komosa, MaÅ‚gorzata DzierzÄ™cka, Tadeusz Mizera, Maciej Gogulski, Anita Zaworska-Zakrzewska.
We appreciate the reviewers very much for the constructive comments on this manuscript. This manuscript has been carefully revised according to these comments on a point-by-point basis. The revised parts of the manuscript was marked red font. The detailed responses to the reviewer’s comments are in the following separate pages. If you have any further questions regarding this manuscript, please let us know.
Thank you very much.
Detailed response to Reviewer
Overall comment:
We are very much thankful to the reviewers for the review. We have revised our present research paper in the light of their useful suggestions and comments. We hope our revision has improved the paper to a level of their satisfaction.
Detailed response to the Reviewers
Reviewer 1:
The purpose of the reviewed work was to analyze and evaluate the comparison of selected mineral elements in tibiotarsal bone one of the biggest avian raptors (White-Tailed Sea Eagle) distributed widely across Eurasian territory. The results of the analysis of the mineral composition of long bones collected from seven White tailed sea eagles are presented and the specimens came from the territories of Western Poland. The authors attempt to show that Pb intoxication may not affect bone mineral composition in raptors, and that assessing the composition of their bones can help determine the the possible cause of their death. The work is interesting, but the authors did not avoid some deficiencies. The determination of the origin of the birds should be placed in the materials and methods, not in the discussion. Moreover, in my opinion, not enough birds were selected for the study. Indeed, determining the mineral composition of bones can be helpful in obtaining more information about birds found dead with an unknown cause, but not in such a small research group. The value of the work would significantly increase if the determinations of elemental content also included organs such as the liver and kidneys. Despite some shortcomings, the topic of the article is adequate to its content. The article is written in a clear and understandable manner. In my opinion, the manuscript can be ready for publication after making the recommended corrections.
Response: Thank you for your valuable comments.
The paragraph on birds’ origin was deleted from the discussion section, some part added in M&M. We are aware that the analysis would be more informative with a bigger group, but obtaining the material from such rare birds is hard, and getting the appropriate amount of those species’ remains may take years. We hope this study serves as a base for further studies on this species. We didn’t have access to soft tissue samples since we were working on birds’ remains. However, we plan to compare the data from this study with more samples we will get in the future.
Thank you very much,
Yours sincerely,
Anita Zaworska-Zakrzewska
Department of Animal Nutrition
Poznań University of Life Sciences, Wołynska 33,
60-637 Poznan, Poland;
E-mail: anita.zaworska-zakrzewska@up.poznan.pl

Reviewer 2 Report
This is an interesting manuscript, and will attract more audience working in this field. , there some typos and mistakes. The units used in the Materials and Methods section and Results section are not consistent. Please fix them.
The authors have developed a new method to evaluate the heavy metal pollution in the environment by analyzing the metal composition in the bones of white-tailed sea eagle. This is very promising and interesting.
This study is original, since the authors chose death raptors found around. Compared with other research using live animal or eggs, this study causes very limited hurt to the animal and environment.This manuscript is well drafted. It is clear and easy to read. The conclusions are well presented, and results could support the conclusion of the manuscript. The proposed questions are also addressed in this study.
Author Response
Dear Editor and Reviewer of Animals,
Thank you for the valuable suggestions and comments made by the editor and reviewers for the improvement of our manuscript entitled “Profile of Selected Mineral Elements in Tibiotarsal bone of the White-Tailed Sea Eagle in its natural habitat ” by Jakub J. Ruszkowski, Anetta Hanć, Marcin Komosa, MaÅ‚gorzata DzierzÄ™cka, Tadeusz Mizera, Maciej Gogulski, Anita Zaworska-Zakrzewska.
We appreciate the reviewers very much for the constructive comments on this manuscript. This manuscript has been carefully revised according to these comments on a point-by-point basis. The revised parts of the manuscript was marked red font. The detailed responses to the reviewer’s comments are in the following separate pages. If you have any further questions regarding this manuscript, please let us know.
Thank you very much.
Detailed response to Reviewer
Overall comment:
We are very much thankful to the reviewers for the review. We have revised our present research paper in the light of their useful suggestions and comments. We hope our revision has improved the paper to a level of their satisfaction.
Detailed response to the Reviewers
Reviewer 2:
This is an interesting manuscript, and will attract more audience working in this field. , there some typos and mistakes. The units used in the Materials and Methods section and Results section are not consistent. Please fix them.
Response:Thanks a lot for your appreciation and recommendation of our manuscript. This manuscript has been carefully revised according to the reviewer’s kind suggestions. Mistakes and typos were corrected. The units in Ca and P values are different (g/kg not µg/kg) to make a table clearer to read. The unit used in every element is provided in the first column of each table. If you have any further questions regarding this manuscript, please let us know.
The authors have developed a new method to evaluate the heavy metal pollution in the environment by analyzing the metal composition in the bones of white-tailed sea eagle. This is very promising and interesting.
This study is original, since the authors chose death raptors found around. Compared with other research using live animal or eggs, this study causes very limited hurt to the animal and environment.
This manuscript is well drafted. It is clear and easy to read. The conclusions are well presented, and results could support the conclusion of the manuscript. The proposed questions are also addressed in this study.
Thank you very much for the kind words about our research.
Yours sincerely,
Anita Zaworska-Zakrzewska
Department of Animal Nutrition
Poznań University of Life Sciences, Wołynska 33,
60-637 Poznan, Poland;
E-mail: anita.zaworska-zakrzewska@up.poznan.pl
